# Efficacy of 1-Kestose Supplementation in Patients with Pancreatic Ductal Adenocarcinoma: A Randomized Controlled Pilot Study

**DOI:** 10.3390/nu16172889

**Published:** 2024-08-29

**Authors:** Kazunori Nakaoka, Eizaburo Ohno, Kento Kuramitsu, Teiji Kuzuya, Kohei Funasaka, Takumi Tochio, Tadashi Fujii, Hideaki Takahashi, Nobuhiro Kondo, Ryoji Miyahara, Senju Hashimoto, Yoshiki Hirooka

**Affiliations:** 1Department of Gastroenterology and Hepatology, Fujita Health University, 1-98 Dengakugakubo, Kutsukake-cho, Toyoake 470-1192, Aichi, Japan; knakaoka@fujita-hu.ac.jp (K.N.); eizaburo.ono@fujita-hu.ac.jp (E.O.); kuramitsu.kento.m3@s.mail.nagoya-u.ac.jp (K.K.); teiji.kuzuya@fujita-hu.ac.jp (T.K.); k-funa@fujita-hu.ac.jp (K.F.); takumitochiobiz@gmail.com (T.T.); fujiitd914@gmail.com (T.F.); h-takahashi@biosislab.co.jp (H.T.); nobuhiro.kondo@itochu-sugar.co.jp (N.K.); myhr@med.nagoya-u.ac.jp (R.M.); 2Department of Applied Biosciences, Graduate School of Bioagricultural Sciences, Nagoya University, Nagoya 464-8601, Aichi, Japan; 3BIOSIS Lab. Co., Ltd., Toyoake 470-1192, Aichi, Japan; 4Department of Medical Research on Prebiotics and Probiotics, Fujita Health University, 1-98 Dengakugakubo, Kutsukake-cho, Toyoake 470-1192, Aichi, Japan; 5Graduate School of Nutritional Sciences, Nagoya University of Arts and Sciences, Nisshin 470-0131, Aichi, Japan; 6Research and Development Division, Itochu Sugar Co., Ltd., Nagoya 447-0834, Aichi, Japan; 7Department of Gastroenterology, Fujita Health University Bantane Hospital, Nagoya 454-8509, Aichi, Japan; hsenju@fujita-hu.ac.jp

**Keywords:** pancreatic cancer, albumin, 1-kestose, NLR, microbiota

## Abstract

Less than half of all patients diagnosed with pancreatic ductal adenocarcinoma (PDAC) respond to chemotherapy, and the prognosis of PDAC is poor, which may be mediated by the gut microbiota. We investigated the clinical improvement effects of 1-kestose, a fructooligosaccharide, on PDAC chemotherapy in this single-center, randomized, controlled pilot trial conducted at Fujita Health University Hospital, which enrolled patients with PDAC. The trial included 1-kestose administration and non-administration groups. The 1-kestose group received 9 g of 1-kestose daily for 12 weeks, and their blood markers, imaging studies, physical findings, and gut microbiota were evaluated. In the 1-kestose administration group, the cancer marker CA19-9 significantly decreased, and there was a reduction in the neutrophil-to-lymphocyte ratio (NLR). There was also suppression of the reduction of albumin levels and of an increase in C-reactive protein. Additionally, *Escherichia coli*, which typically increases in PDAC, significantly decreased in the 1-kestose group. Thus, 1-kestose altered the gut microbiota and improved the prognostic factors for PDAC. Large-scale, long-term trials of 1-kestose interventions for PDAC are thus warranted to improve the prognosis of PDAC.

## 1. Introduction

Pancreatic ductal adenocarcinoma (PDAC) is mostly unresectable or metastatic and is estimated to become a leading cause of cancer-related deaths by 2030 [1,2,3]. PDAC is one of the most malignant tumors, with an extremely low 5-year survival rate (<10%) [4,5], and can lead to pancreatic exocrine insufficiency-related nutrient malabsorption, which may contribute to its poor prognosis [6]. Although chemotherapy is the first-line treatment for PDAC, fewer than half of all patients with PDAC respond to it; this lack of response cannot be explained solely by genetic factors [7]. Thus, environmental factors, including the gut microbiome, may mediate the response to chemotherapy [8,9].

In recent years, various cohort studies have shown that the gut microbiota of patients with pancreatic cancer significantly differs from that of healthy individuals and that dysbiosis of the gut microbiota may affect the efficacy of chemotherapy and be associated with the prognosis of PDAC [10,11]. Particularly in Japan, patients with PDAC had significantly increased levels of oral bacteria in their gut as well as decreased levels of *Anaerostipes* and propionic acid [12]. As short-chain fatty acids (SCFAs), including propionic acid, regulate immune function via regulatory T cells, SCFAs may be involved in the effectiveness of chemotherapy [13].

Although several signs of progress have been seen in the treatment of pancreatic cancer [14,15,16,17], clinical trial data are still largely missing and are urgently needed to verify the relevant effects and for the development of more personalized treatment approaches. 1-Kestose is a fructooligosaccharide, which is composed of fructose and sucrose, and it increases *Bifidobacterium* and *Anaerostipes* species, as well as the amount of SCFAs, in the gut [18]. Recent reports indicate that 1-kestose improves symptoms in patients with ulcerative colitis and enhances the nutritional status of patients with sarcopenia [19,20]. However, the efficacy of 1-kestose in improving the chemotherapy outcomes and nutritional status in patients with PDAC has not yet been investigated. We posited that the oral intake of 1-kestose could modulate the gut microbiota of patients with unresectable PDAC and thereby improve the nutritional status and antitumor effects. Therefore, to preliminarily elucidate the relationship between chemotherapy and the gut environment in PDAC, we conducted a pilot study to evaluate the effects of 1-kestose administration on improving the nutritional status and treatment outcomes in PDAC patients, as well as to investigate the changes in the gut microbiota.

## 2. Materials and Methods

### 2.1. Study Design and Participants

This single-center, randomized, open-label pilot study comprised 1-kestose administration and non-administration groups of patients with PDAC at Fujita Health University Hospital, Aichi, Japan. Forty patients with untreated unresectable PDAC were enrolled between August 2021 and September 2023 at Fujita Health University Hospital. The inclusion criteria were as follows. (1) The study’s purpose and procedures were thoroughly explained to potential participants, who then provided written informed consent to participate in the study. (2) Participants were between 20 and 80 years of age at the time that they provided informed consent to participate in the study. (3) PDAC was diagnosed based on contrast-enhanced computed tomography findings, with adenocarcinoma detected on histological examination of an endoscopic ultrasound-guided fine-needle biopsy specimen. (4) Patients with an ECOG Performance Status (PS) of 0–1 and (5) those who were not receiving antibiotics, prebiotics, or probiotics, and who had no change in treatment in the 4 weeks before study initiation, were eligible for inclusion in the study. The exclusion criteria were as follows: (1) a history of gastrointestinal surgery, radiation therapy, or chemotherapy; (2) a history of biliary drainage for obstructive jaundice; and (3) severe hepatic, renal, or cardiovascular diseases. In principle, the standard GnP was used for chemotherapy [7], although a modified FOLFIRINOX regimen [21] was used in one patient in each study arm. Table 1 shows the pretreatment characteristics of both groups.

The study protocol was approved by the research ethics committee of Fujita Health University (CR23-002) and was registered in the Japan Registry of Clinical Trials (jRCTs: https://jrct.niph.go.jp, accessed on 10 March 2021); the trial number is jRCTs041200109. The trial was conducted in accordance with the guidelines of the Declaration of Helsinki. All the participants provided written informed consent to participation in the study.

### 2.2. Randomization and Masking

Forty patients who were diagnosed with PDAC were enrolled and randomly assigned 1:1 to the 1-kestose administration or the non-administration group. Stratification was performed according to: (1) the pretreatment albumin levels (≥40 g/L vs. <40 g/L) and (2) the age (≥65 years vs. <65 years) just before the first treatment. Although the 1-kestose product is individually packaged in opaque medicine bags, wherein each contains a single dose, which makes it impossible to discern the contents merely by appearance, this pilot study is part of an open-label administration study where the product was administered only to cases in the 1-kestose administration group.

### 2.3. Procedures

Patients in the 1-kestose group received a dose of 9 g/day 1-kestose powder containing 98% 1-kestose (Itochu Sugar Co., Ltd, Hekinan, Japan, 3.0 g per dose, three times daily) for 12 weeks. The 1-kestose dose was determined in accordance with the maximum permissible dose. During the trial period, the patients continued any regular treatment they were receiving and were instructed to maintain their preintervention diet and exercise habits. The baseline characteristics, medical history, and current medications of each patient were recorded during the screening visit.

The nutritional status and effectiveness of chemotherapy were determined at weeks 0 and 12 by hematological examination, including of the neutrophil and lymphocyte counts and levels of albumin, C-reactive protein (CRP), and CA19-9 to assess the degree of response according to the revised RECIST guidelines for contrast-enhanced computed tomography [22].

Fecal samples were collected at weeks 0 and 12 to evaluate the gut microbiome. Adverse events were recorded, and study participation was discontinued if the patient required additional treatment for disease exacerbation or complications, or if the participant wished to withdraw consent.

### 2.4. Preparation of Fecal Samples

At baseline and approximately 12 weeks later, fecal samples from the PDAC patients were collected using the Fecal Collection Kit FS-0017 (Techno Suruga Laboratory, Shizuoka, Japan). Fecal DNA was extracted using the QIAamp PowerFecal Pro DNA Kit (QIAGEN, Hilden, Germany) in accordance with the manufacturer’s instructions. Furthermore, the study was approved by the Research and Ethics Committee of the Fujita Health University (approval no. HM22-272 and HM23-078), registered in the University Hospital Medical Information Network Clinical Trials Registry (UMIN000054817), and conducted in accordance with the Declaration of Helsinki (1975).

### 2.5. Analysis of the Fecal Microbiome

The 16S rRNA amplicons from the fecal DNA samples from the PDAC patients were generated using the 341F/785R primers recommended by Illumina [23] and TaKaRa Ex Taq Hot Start Version (RR006A, Takara Bio Inc., Shiga, Japan). The PCR conditions were as follows: an initial denaturation at 94 °C for 2 min, followed by 30 cycles of denaturation at 94 °C for 30 s, annealing at 55 °C for 30 s, and extension at 72 °C for 30 s, concluding with a final extension at 72 °C for 5 min. Dual indices were then attached, and 16S rRNA next-generation sequencing (NGS) was performed by Bioengineering Lab Co., Ltd. (Fukushima, Japan). The sequencing data were processed, analyzed, and visualized using the EzBioCloud 16S database and 16S microbiome pipeline by ChunLab Inc. (EzBioCloud 16S-based MTP app, available at https://www.EZbiocloud.net, accessed on 10 March 2021). The EzBioCloud 16S database was designed for species-level identification, although there are limitations owing to the lack of sequence differences in some closely related species. The combination of the database and bioinformatics pipelines allows species-level exploration of microbiome data [24]. The NGS dataset of 85 healthy controls, enrolled in another study and deposited in the NCBI Sequence Read Archive under accession number PRJNA1075329, was used to compare the microbiomes with those of the PDAC patients. To evaluate the richness and evenness of the microbial samples, we utilized Chao1 estimation and the Shannon Diversity Index for the alpha diversity. The beta diversity, which indicates the phylogenetic distance among groups, was assessed using the Generalized UniFrac distance and visualized by principal coordinate analysis (PCoA). Intergroup differences in the alpha and beta diversities were tested using the Wilcoxon rank-sum test and permutational multivariate analysis of variance (PERMANOVA) with 9999 permutations, respectively. The species distinguishing each group were identified, based on a linear discriminant analysis (LDA) score greater than 4.0, by using the Linear Discriminant Analysis Effect Size (LEfSe) algorithm to compare the healthy group to the PDAC patients in both the 1-kestose administration group and the non-administration group and with significance set at a *p*-value of <0.05.

### 2.6. Bioelectrical Impedance Analysis

The body composition parameters included the body mass index (BMI), extracellular water to total body water ratio (ECW:TBW), and whole-body phase angle (PhA), which were measured at weeks 0 and 12 of the intervention using direct segmental multifrequency bioelectrical impedance analysis with an InBody 770 device (InBody Inc., Tokyo, Japan). The participants maintained a standing posture for 15 s during the measurement. The PhA value at a frequency of 50 kHz was used, and this was measured while maintaining a standing posture for 15 s, and calculated as the whole-body PhA.

### 2.7. Statistical Analysis

Owing to the lack of previous reports on the effect of 1-kestose on PDAC, an accurate power analysis calculation for the sample size was not performed in this pilot trial. Continuous variables were expressed as the mean (SD), and categorical data as the frequency (proportion). Differences between paired samples were analyzed using the Wilcoxon signed-rank test. Intergroup differences were assessed using the Mann–Whitney U test for continuous data and Fisher’s exact test for categorical data, as appropriate. All the analyses were performed using GraphPad Prism (version 10.2, GraphPad Software, La Jolla, CA, USA). Statistical significance was set at *p* < 0.05.

## 3. Results

### 3.1. Participant Selection and Clinical Characteristics

Figure 1 comprises a flowchart that illustrates the participant selection as per the study protocol. All of the 40 patients with PDAC who were assessed for eligibility were enrolled and evenly allocated to the two groups: 20 each in the 1-kestose administration and non-administration groups. Two patients discontinued the study because of worsening conditions that prevented regular hospital visits. Consequently, 38 patients completed the treatment: 20 in the 1-kestose administration group and 18 in the non-administration group. Table 1 presents the baseline characteristics of the two groups, which showed no significant intergroup differences.

### 3.2. Endpoints

We investigated the effects of 1-kestose intake on the nutritional status of patients with PDAC. The albumin levels decreased from weeks 0 to 12 in the non-administration group (*p* = 0.001) but not in the 1-kestose administration group (*p* = 0.537; Figure 2A). From weeks 0 to 12, there was no significant intergroup difference in the change in the BMI (Figure 2B). The ECW:TBW significantly increased in both groups from weeks 0 to 12 (Figure 2C). The PhA significantly decreased in both groups from weeks 0 to 12 (*p* < 0.001; Figure 2D).

Next, we investigated the effects of 1-kestose intake on the antitumor effects in patients with PDAC. The neutrophil count at week 12 was lower in the 1-kestose group than in the non-administration group (*p* = 0.013; Figure 3A). At week 12, compared to the non-administration group, the 1-kestose group had a higher lymphocyte count (*p* = 0.033; Figure 3B) and lower neutrophil-to-lymphocyte ratio (NLR; *p* = 0.026; Figure 3C).

From weeks 0 to 12, the CRP levels increased significantly in the non-administration group (*p* = 0.011) but not in the 1-kestose administration group (*p* = 0.327; Figure 3D), whereas the CA19-9 level decreased significantly in the 1-kestose administration group (*p* < 0.001) but not in the non-administration group (*p* = 0.144) (Figure 3E).

In the non-administration and 1-kestose administration groups, according to the RECIST classification, a response rate (complete response or partial response) was recorded in 11 (61%) and 14 (70%) patients, respectively, and a disease control rate (defined at 12 weeks, which confirmed the complete response, partial response, or stable disease) in 15 (83%) and 19 (95%) patients, respectively, and both showed no significant intergroup difference (Table 2).

### 3.3. Fecal Microbiome

First, we compared the differences in the gut microbiota between healthy individuals and patients with PDAC. The Chao1 and Shannon indices, which indicate the alpha diversity, were higher in the PDAC group than in the healthy group (Figure 4A,B). The beta diversity, as measured by the Generalized UniFrac distance, showed significant intergroup differences (*p* < 0.001; Figure 4C). Next, we conducted LEfSe analysis to identify the species-level intergroup differences in the gut microbiota (Table 3). Compared to those in the healthy group, participants in the PDAC group had increased proportions of *Escherichia coli*, *Streptococcus salivarius*, and *Enterococcus faecium* as well as a decreased proportions of *Blautia wexlerae* and *Bifidobacterium faecium*.

Next, we compared the gut microbiota of the 1-kestose administration group and the non-administration group. The Chao1 index, which represents the alpha diversity, significantly decreased at week 12 compared to week 0 in both groups, whereas the Shannon Diversity Index significantly decreased in only the non-administration group (Figure 5A,B). The beta diversity, as measured by the Generalized UniFrac distance, showed no significant differences between baseline and post-intervention (Figure 5C).

Subsequently, we compared the gut microbiota before and after the intervention in the non-administration and 1-kestose administration groups, focusing on bacteria that showed significant changes in patients with PDAC as compared to healthy individuals (Table 4). The abundance of *E. coli*, which was abundant in the PDAC group, significantly decreased at week 12 in the 1-kestose administration group but showed no significant change in the non-administration group. Before and after treatment, there was no significant difference in the *Bifidobacterium*, *Anaerostipes*, and *Faecalibacterium* groups (Appendix A), although these have been previously reported to increase with 1-kestose administration.

### 3.4. Safety Assessment

Two patients in the non-administration group experienced worsening of their condition owing to the primary disease, which made it impossible to continue chemotherapy, and they were switched to palliative care (considered sensor cases). No serious adverse event related to 1-kestose or chemotherapy was observed in either group.

## 4. Discussion

Owing to pancreatic exocrine insufficiency, PDAC significantly alters the nutritional status of patients [25]. Additionally, the majority of PDAC cases have sarcopenia, which is considered one of the prognostic factors for PDAC. Therefore, nutritional intervention for PDAC may improve the prognosis [26,27,28]. In this study, to evaluate the changes in nutritional status with 1-kestose intake, we used the ECW:TBM and PhA, which are indicators of nutritional status and sarcopenia, respectively. Both the 1-kestose administration group and the non-administration group showed a significant increase in the ECW:TBW and a significant decrease in PhA from weeks 0 to 12. Although high PhA levels could improve the prognosis in PDAC [29], the results of this study indicated that 1-kestose intake does not improve PhA. Moreover, the blood albumin concentration is a nutritional marker, and decreased albumin levels are associated with poor prognosis in PDAC [30,31]. In this study, the blood albumin significantly decreased from weeks 0 to 12 in the 1-kestose non-administration group, whereas no significant change was observed in the 1-kestose administration group. These results suggest that 1-kestose may partially improve the nutritional status of PDAC patients.

Next, we found that the intake of 1-kestose significantly reduced the expression of the cancer marker CA19-9 in patients who were undergoing chemotherapy for PDAC. Recently, the NLR has garnered attention as a prognostic factor for PDAC. The NLR fluctuates in various diseases, and a value exceeding 3.0 in adults suggests a potential underlying disease [32]. Additionally, because the NLR increases in PDAC, it is being highlighted as a biomarker for PDAC [33]. In this trial, the baseline NLR of the patients in both groups exceeded the standard value of 3.0 (Table 1). The NLR is considered a prognostic factor for PDAC because a lower NLR is associated with increased overall survival (OS), 5-year OS, and disease-free survival (DFS), along with a reduction in the tumor size and CA19-9 levels [34,35,36,37,38]. In this study, the 1-kestose intake may have decreased the CA19-9 levels by reducing the neutrophils, increasing the lymphocytes, and consequently lowering the NLR. However, 1-kestose intake did not result in a significant intergroup difference in the response rate.

The CRP levels constitute important, promising inflammatory prognostic factors in pancreatic cancer [39,40]. CRP is used as an inflammatory marker and is related to the gut microbiota [41]. In this study, 1-kestose suppressed the increase in the CRP levels, which suggests an improvement in the inflammatory state.

In this study, we compared the gut microbiota of healthy individuals and patients with PDAC and found a significant increase in *E. coli* in the PDAC group. Several reports have indicated that *E. coli* is involved in the onset and progression of pancreatic cancer [42,43,44]. Some *E. coli* strains contribute to neutrophil activation and inhibit lymphocyte activation [45,46]. Additionally, *E. coli* disrupts the intestinal barrier function, potentially increases the circulation of lipopolysaccharides in the blood owing to enhanced intestinal permeability, and thereby induces an inflammatory state [47]. In this study, therapeutic intervention with 1-kestose significantly reduced *E. coli*, decreased the NLR, suppressed the increase in the CRP levels, and prevented the decrease in the albumin levels. Therefore, *E. coli* may be an important bacterial species that determines the prognosis, inflammation, and nutritional status of patients with pancreatic cancer. However, further studies using molecular biological methods are necessary to elucidate the detailed mechanisms that are involved.

Intervention trials with 1-kestose in humans and rats reported an increase in the *Bifidobacterium*, *Anaerostipes*, and *Faecalibacterium* species. However, this study did not confirm the changes in the occupancy rates of these bacteria [48,49]. In studies involving 1-kestose intervention in patients with ulcerative colitis, an increase in these bacteria was not observed; however, an increase in *Erysipelotrichaceae* was noted. The results of this trial and these reports suggest that 1-kestose might form a different gut microbiota in patients with various diseases compared to under normal conditions [19]. Furthermore, in a report on 1-kestose intervention in patients with sarcopenia, an increase in *Bifidobacterium* was observed between weeks 4 and 8 of the intervention, followed by a decrease at 12 weeks. This suggests that the effects of 1-kestose on the gut microbiota might be temporary [20]. Moreover, as no significant changes were observed in the SCFA-producing bacteria, it is unlikely that they influenced chemotherapy through Tregs-based mechanisms. Further investigation is necessary to elucidate the detailed mechanisms.

This study suggested that 1-kestose intake improved the antitumor effects of PDAC chemotherapy and nutritional status, although the underlying mechanisms remain unclear. Given that the antitumor effects of immune checkpoint inhibitors have been shown to be enhanced by *Bifidobacterium* administration [50], it is possible that alterations in the gut microbiota may have influenced the chemotherapy efficacy in this study as well. Additionally, as short-chain fatty acids, which increase with 1-kestose intake, have been reported to improve metabolic function [51] and exhibit anti-inflammatory effects through G-protein-coupled receptor 43 [41], it is plausible that specific metabolites impacted the albumin and CRP levels in the blood in this study. Further investigation is required to elucidate the detailed mechanisms.

Our study had several limitations. First, the sample size was small, with only 38 Japanese participants, and the absence of a placebo group may have introduced bias into the results. Future studies should involve a larger sample size and include a placebo group to ensure more reliable findings. Second, the observation period was short (12 weeks), which allowed us to verify only PDAC’s prognostic factors, such as NLR. Since 1-kestose has been reported to improve insulin sensitivity after 12 weeks of intake [52], and as gemcitabine is clinically recommended to be evaluated for efficacy every 12 weeks [7], the study period was also set to 12 weeks. Yet long-term observations are required to elucidate the effects of 1-kestose on PDAC prognosis. Third, we did not measure the gut microbiota metabolites, such as SCFA, in the feces and blood, and this results in many uncertainties regarding the causal relationship between changes in the gut microbiota and host phenotypes. Fourth, it is unclear whether the effects observed in this trial are solely due to 1-kestose or occurred in the context of its combination with chemotherapy, and it is necessary to ascertain this in further research.

## 5. Conclusions

In conclusion, our study identified a novel therapeutic approach that improves PDAC’s prognostic factors and patients’ nutritional status through the oral intake of 1-kestose during chemotherapy for PDAC. The fact that 1-kestose is present in onions and rye and that crystallized products are available makes dietary intervention highly appealing. Furthermore, improving the nutritional status with 1-kestose may significantly contribute to the prognosis and quality of life of PDAC patients. However, as this was a pilot study with a small sample size, larger clinical trials and a detailed elucidation of the underlying mechanisms are required.

## Figures and Tables

**Figure 1 nutrients-16-02889-f001:**
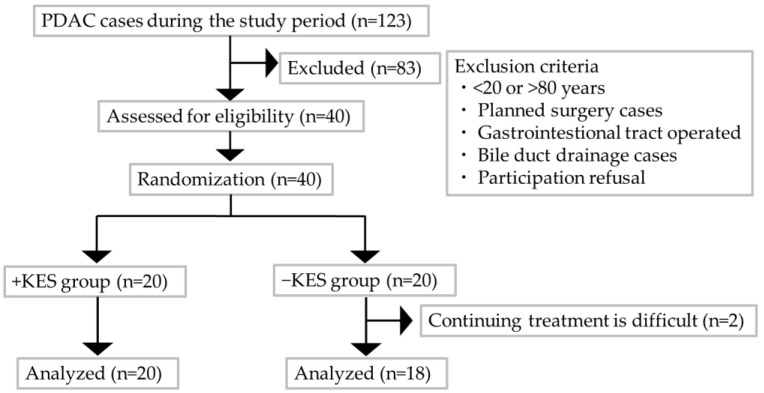
Flowchart of the study protocol for patient selection. −KES, group not administered 1-kestose; +KES, group administered 1-kestose.

**Figure 2 nutrients-16-02889-f002:**
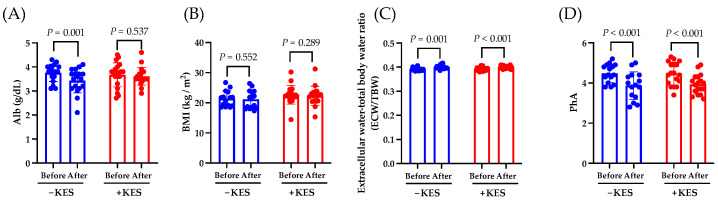
Analysis of the endpoints. Data are presented as the mean ± SD. (**A**) Change in albumin (Alb), (**B**) change in BMI, (**C**) change in extracellular water–total body water ratio, and (**D**) change in PhA from baseline to week 12. The Wilcoxon test was used for calculating statistical significance. −KES, group not administered 1-kestose; +KES, group administered 1-kestose.

**Figure 3 nutrients-16-02889-f003:**
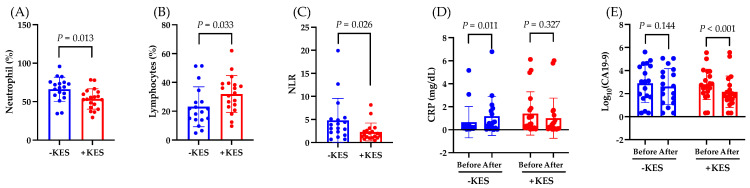
Analysis of the endpoints. Data are presented as the mean ± SD. (**A**) Neutrophil count (%), (**B**) lymphocyte count (%), and (**C**) neutrophil to lymphocyte ratio (NLR) at week 12. The Mann–Whitney U test was used to calculate statistical significance. (**D**) Change in C-reactive protein (CRP), and (**E**) change in CA19-9 from baseline to week 12. The Wilcoxon test was used for calculating statistical significance. −KES, group not administered 1-kestose; +KES, group administered 1-kestose.

**Figure 4 nutrients-16-02889-f004:**
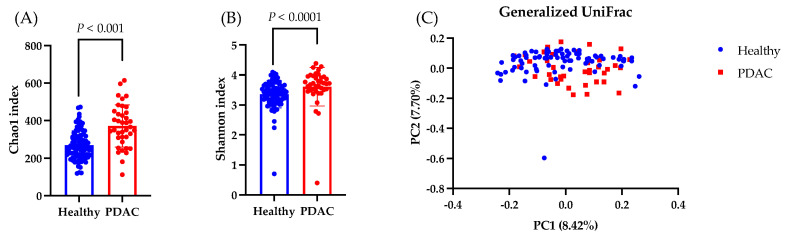
Analysis of the gut microbiota. Data are presented as the mean ± SD. (**A**,**B**) Comparing the observed Chao1 and Shannon indices as measures of the alpha diversity in the healthy and PDAC groups. The Wilcoxon test was utilized to calculate statistical significance. (**C**) The differential beta diversity of the gut microbiota was examined in Generalized UniFrac. Multivariate analysis of variance with permutational multivariate analysis of variance, *p* = 0.001.

**Figure 5 nutrients-16-02889-f005:**
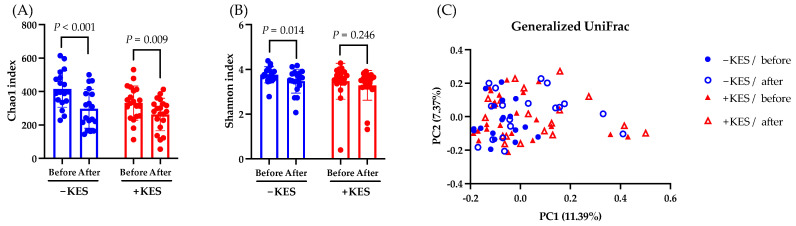
Analysis of the gut microbiota. Data are presented as the mean ± SD. (**A**,**B**) Comparing the observed Chao1 and Shannon indices as measures of the alpha diversity from baseline to week 12 in the −KES and +KES groups. The Wilcoxon test was utilized to calculate statistical significance. (**C**) The differential beta diversity of the gut microbiota was examined in Generalized UniFrac. Multivariate analysis using permutational multivariate analysis of variance showed a significant difference between −KES/before and +KES/after at *p* = 0.001 but not between the other groups. −KES, group not administered 1-kestose; +KES, group administered 1-kestose.

**Table 1 nutrients-16-02889-t001:** Baseline characteristics of participants (*n* = 38) included in the analysis.

Characteristic	−KES	+KES	*p* Value
Sex (male/female)	8/10	10/10	0.757
ECOG (0/1)	16/2	18/2	>0.999
Stage (III/IV)	9/9	10/10	>0.999
Age at entry, mean ± SD	67.50 ± 9.40	68.95 ± 9.34	0.637
BMI (kg/m^2^), mean ± SD	21.37 ± 2.59	22.68 ± 3.14	0.069
Extracellular water–total body water ratio (ECW/TBW), mean ± SD	0.39 ± 0.01	0.39 ± 0.01	0.512
PhA, mean ± SD	4.48 ± 0.46	4.46 ± 0.57	0.922
Alb (g/dL), mean ± SD	3.76 ± 0.36	3.67 ± 0.50	0.525
CRP (mg/dL), mean ± SD	0.62 ± 1.33	1.43 ± 1.88	0.080
Neutrophil (%), mean ± SD	68.29 ± 10.15	69.48 ± 12.34	0.767
Lymphocytes (%), mean ± SD	23.31 ± 8.88	21.38 ± 9.13	0.857
NLR, mean ± SD	3.48 ± 1.73	5.28 ± 7.17	0.989
CA19-9 (Log10(CA19-9)), mean ± SD	2.89 ± 1.66	2.81 ± 1.31	0.874

−KES, the group that did not receive 1-kestose; +KES, the group that was administered 1-kestose.

**Table 2 nutrients-16-02889-t002:** Analysis of the endpoints by using the Evaluation Criteria in Solid Tumors (RECIST).

Characteristic	−KES	+KES
Complete response	0	0
Partial response	11 (61%)	14 (70%)
Stable disease	4 (22%)	5 (25%)
Progressive disease	3 (17%)	1 (5%)

−KES, the group that did not receive 1-kestose; +KES, the group that was administered 1-kestose.

**Table 3 nutrients-16-02889-t003:** Gut microbiota analysis.

Taxon Name	LDA Effect Size	*p* Value	Taxonomic Relative Abundance	
			Healthy	PDAC
*Escherichia coli* group	4.228	0.013	0.741	3.765
*Blautia wexlerae*	4.207	0.001	6.002	2.968
*Streptococcus salivarius* group	4.176	0.001	0.499	3.547
*Bifidobacterium adolescentis* group	4.119	0.044	3.577	1.185
*Enterococcus faecium* group	4.029	0.015	1.023	2.550

Species-level comparison of the microbiome in the feces of the healthy and PDAC groups using linear discriminant analysis (LDA) effect size; LDA score > 4.

**Table 4 nutrients-16-02889-t004:** Analysis of the gut microbiota.

Taxon Name	Taxonomic Relative Abundance					
	−KES			+KES		
	Before	After	*p*-Value	Before	After	*p*-Value
*Escherichia coli* group	3.74 ± 10.08	4.28 ± 5.98	0.799	3.78 ± 9.33	1.76 ± 4.08	0.023
*Blautia wexlerae*	3.08 ± 3.94	3.45 ± 3.68	0.832	2.87 ± 2.20	2.67 ± 2.73	0.523
*Streptococcus salivarius* group	3.13 ± 5.95	3.97 ± 6.55	0.417	3.92 ± 6.40	5.03 ± 5.03	0.368
*Bifidobacterium adolescentis* group	1.62 ± 3.26	3.02 ± 9.79	0.570	0.80 ± 2.44	1.53 ± 4.27	0.722
*Enterococcus faecium* group	0.02 ± 0.03	0.85 ± 3.10	0.079	4.83 ± 20.39	4.10 ± 12.90	0.540

−KES, the group that did not receive 1-kestose; +KES, the group that was administered 1-kestose. Comparison of the microbiome in feces from baseline to week 12 in −KES and +KES group using the Wilcoxon test.

## Data Availability

The data from this study are available upon request from the corresponding author.

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
