# Peer review of "Efficacy of 1-Kestose Supplementation in Patients with Pancreatic Ductal Adenocarcinoma: A Randomized Controlled Pilot Study"

_nutrients, 2024, doi:10.3390/nu16172889_

Round 1

Reviewer 1 Report

Comments and Suggestions for Authors

The manuscript presents a well-conducted pilot study with promising findings on the potential benefits of 1-kestose in PDAC treatment. With some improvements, especially in providing a broader context, discussing mechanisms in more detail, and justifying methodological choices, the manuscript can significantly contribute to the field.

-        INTRODUCTION: The background is not sufficient and has critical flaws. The readers may benefit from some more background and context. For example, a short introduction on the ongoing preclinical efforts to identify novel targets and to develop novel strategies to treat pancreatic cancer would provide some orientation to readers and set the scene. For this purpose, I suggest seeing the following paper and adding the following sentence on page 2 line 53: “Although several signs of progress were made in the treatment of pancreatic cancer [1-4], clinical trial data is still largely missing and is urgently needed to verify relevant effects and for the development of more personalized treatment approaches.

1.     Mar Drugs. 2023 May 4;21(5):288. doi: 10.3390/md21050288.

2.     Cancers (Basel). 2023 Jun 30;15(13):3423. doi: 10.3390/cancers15133423.

3.     Cancers (Basel). 2020 Oct 31;12(11):3206. doi: 10.3390/cancers12113206. 

4.      Mar Drugs. 2023 Jul 19;21(7):412. doi: 10.3390/md21070412. 

-        METHODS: Please Justify the choice of the 12-week observation period, especially given the short-term nature mentioned in the discussion. Moreover, consider including a flowchart for participant recruitment and follow-up for better clarity.

-        RESULTS: Include more detailed baseline characteristics of the participants to understand any initial differences between groups.

-        DISCUSSION: Expand on the discussion of the mechanisms by which 1-kestose might affect chemotherapy outcomes, possibly incorporating more recent studies.

-        CONCLUSION: Strengthen the conclusion by explicitly stating how these findings could influence clinical practice if validated by larger studies.

-        Minor grammatical errors and typos should be corrected for better readability.

Comments on the Quality of English Language

Minor editing of English language required.

Author Response

Reviewer1

- INTRODUCTION: The background is not sufficient and has critical flaws. The readers may benefit from some more background and context. For example, a short introduction on the ongoing preclinical efforts to identify novel targets and to develop novel strategies to treat pancreatic cancer would provide some orientation to readers and set the scene. For this purpose, I suggest seeing the following paper and adding the following sentence on page 2 line 53: “Although several signs of progress were made in the treatment of pancreatic cancer [1-4], clinical trial data is still largely missing and is urgently needed to verify relevant effects and for the development of more personalized treatment approaches.

  1. Mar Drugs. 2023 May 4;21(5):288. doi: 10.3390/md21050288.
  2. Cancers (Basel). 2023 Jun 30;15(13):3423. doi: 10.3390/cancers15133423.
  3. Cancers (Basel). 2020 Oct 31;12(11):3206. doi: 10.3390/cancers12113206. 
  4. Mar Drugs. 2023 Jul 19;21(7):412. doi: 10.3390/md21070412. 

Reply: Thank you for your comment regarding the background in the INTRODUCTION. We have added a brief introduction about the treatment of PDCA and also cited relevant literature.

(L53) Although several signs of progress were made in the treatment of pancreatic cancer [1-4], clinical trial data is still largely missing and is urgently needed to verify relevant effects and for the development of more personalized treatment approaches.

METHODS: Please Justify the choice of the 12-week observation period, especially given the short-term nature mentioned in the discussion.

Reply: Thank you for your comment regarding the observation period. Firstly, there have been reports indicating that 1-kestose improves insulin sensitivity in obese individuals after 12 weeks of intake, suggesting that 12 weeks is sufficient to observe its effects as a prebiotic (https://www.ncbi.nlm.nih.gov/pmc/articles/PMC8470827/). Additionally, based on the large-scale MPACT trial, which demonstrated that the median progression-free survival with gemcitabine is 3.7 months, the 12-week period was clinically adopted as it secures enough time for the treatment to show effects, while also allowing for an early change in treatment strategy if the effects are insufficient (https://pubmed.ncbi.nlm.nih.gov/24131140/). With this in mind, we have added the following sentence.

(L352) Since 1-kestose has been reported to improve insulin sensitivity after 12 weeks of intake, and gemcitabine is clinically recommended to be evaluated for efficacy every 12 weeks, the study period was also set to 12 weeks.

Moreover, consider including a flowchart for participant recruitment and follow-up for better clarity.

Reply: Thank you for your comment regarding the flowchart for participant recruitment and follow-up. As per your suggestion, we have revised Figure 1 to improve clarity.

RESULTS: Include more detailed baseline characteristics of the participants to understand any initial differences between groups.
Reply: Thank you for your comment regarding the baseline characteristics of the participants. As per your suggestion, we have provided a detailed description of the participants' baseline characteristics, which have been included in Table 1. In doing so, we referred to the following paper (https://www.mdpi.com/2072-6694/16/15/2734).

DISCUSSION: Expand on the discussion of the mechanisms by which 1-kestose might affect chemotherapy outcomes, possibly incorporating more recent studies.

Reply: Thank you for your comment regarding the mechanisms by which 1-kestose might affect chemotherapy outcomes. To the best of our knowledge, the mechanisms by which prebiotics enhance the therapeutic effects of chemotherapy for PDAC remain unclear. However, it has been previously reported that Bifidobacterium enhances the antitumor effects of immune checkpoint inhibitors, suggesting that the gut microbiota may also play a role in this study. Nevertheless, the detailed mechanisms need further investigation. Therefore, we have added the following sentence.

(L338) The study suggested that 1-kestose intake improved the anti-tumor effects of PDAC chemotherapy and nutritional status, though the underlying mechanisms remain unclear. Given that the antitumor effects of immune checkpoint inhibitors have been shown to be enhanced by Bifidobacterium administration, it is possible that alterations in the gut microbiota may have influenced chemotherapy efficacy in this study as well. Additionally, as short-chain fatty acids, which increase with 1-kestose intake, have been reported to improve metabolic function and exhibit anti-inflammatory effects through G-protein-coupled receptor 43, it is plausible that specific metabolites impacted albumin and CRP levels in the blood in this study. Further investigation is required to elucidate the detailed mechanisms.

CONCLUSION: Strengthen the conclusion by explicitly stating how these findings could influence clinical practice if validated by larger studies.

Reply: Thank you for your comment regarding the influence on clinical practice. We believe that if larger studies confirm the effects of 1-kestose on PDAC chemotherapy, improvements in nutritional status could lead to better prognosis and enhanced patient quality of life. Therefore, taking your suggestion into consideration, we have added the following sentence.

(L366) Furthermore, improving the nutritional status with 1-kestose may significantly contribute to the prognosis and quality of life of PDAC patients.

Minor grammatical errors and typos should be corrected for better readability.

Reply: Thank you for your comment regarding the minor grammatical errors. I have revised the following section.

(L11) [email protected]
(L15)
[email protected]

・I have revised the references.

Reviewer 2 Report

Comments and Suggestions for Authors

This is a very promising study concerning oral administration of 1-kestose in patients with pancreatic ductal adenocarcinoma. Introduction justifies the study, methods which were applied are appropriate, however I don't understand why control group did not take a placebo for example maltodextrin. We need placebo to avoid placebo effect. I believe it is one of the limitation of your study and it should be mentioned. Besides, you need also to explain how kestose was recommended to consume.

Results are clearly presented and fair discussed, however it could be improved by trying to interpret why kestose patients had different blood albumin or CRP levels. The authors recognize the limitations of their work and indicate future directions of the study.

Author Response

Reviewre2

This is a very promising study concerning oral administration of 1-kestose in patients with pancreatic ductal adenocarcinoma. Introduction justifies the study, methods which were applied are appropriate, however I don't understand why control group did not take a placebo for example maltodextrin. We need placebo to avoid placebo effect. I believe it is one of the limitation of your study and it should be mentioned.

Reply: Thank you for your comment regarding a placebo. The use of prebiotics as an intervention for diseases, including PDAC, remains under debate, particularly concerning the selection of an appropriate placebo. As a result, it was difficult to establish a suitable placebo for this study, and thus we did not include one. However, as you pointed out, the absence of a placebo group may introduce potential bias, so we have revised the following sentence accordingly.

(L348) First, the sample size was small, with only 38 Japanese participants, and the absence of a placebo group may have introduced bias into the results. Future studies should involve a larger sample size and include a placebo group to ensure more reliable findings

Besides, you need also to explain how kestose was recommended to consume.

Reply: Thank you for your comment regarding the consumption of 1-kestose. We have added the following sentence.
(L106) 3.0 g per dose, three times daily

Results are clearly presented and fair discussed, however it could be improved by trying to interpret why kestose patients had different blood albumin or CRP levels. The authors recognize the limitations of their work and indicate future directions of the study.

Reply: Thank you for your comment regarding why 1-kestose patients had different blood albumin or CRP levels. The mechanisms by which 1-kestose affects blood albumin or CRP levels remain unclear. Short-chain fatty acids, which increase with 1-kestose intake, have been reported to improve metabolic function and exert anti-inflammatory effects through G-protein-coupled receptor 43. However, since this is speculative in our study, we believe this should be discussed with caution. Therefore, we have added the following sentence.

(L338) The study suggested that 1-kestose intake improved the anti-tumor effects of PDAC chemotherapy and nutritional status, though the underlying mechanisms remain unclear. Given that the antitumor effects of immune checkpoint inhibitors have been shown to be enhanced by Bifidobacterium administration, it is possible that alterations in the gut microbiota may have influenced chemotherapy efficacy in this study as well. Additionally, as short-chain fatty acids, which increase with 1-kestose intake, have been reported to improve metabolic function and exhibit anti-inflammatory effects through G-protein-coupled receptor 43, it is plausible that specific metabolites impacted albumin and CRP levels in the blood in this study. Further investigation is required to elucidate the detailed mechanisms.

Round 2

Reviewer 1 Report

Comments and Suggestions for Authors

The manuscript is now suitable for the publication